# Y-Chromosome Haplotype Report among Eight Italian Horse Breeds

**DOI:** 10.3390/genes14081602

**Published:** 2023-08-09

**Authors:** Andrea Giontella, Irene Cardinali, Francesca Maria Sarti, Maurizio Silvestrelli, Hovirag Lancioni

**Affiliations:** 1Department of Veterinary Medicine, University of Perugia, 06126 Perugia, Italy; andrea.giontella@unipg.it (A.G.); maurizio.silvestrelli@unipg.it (M.S.); 2Department of Chemistry, Biology and Biotechnology, University of Perugia, 06123 Perugia, Italy; 3Department of Agricultural, Food and Environmental Sciences, University of Perugia, 06121 Perugia, Italy; francesca.sarti@unipg.it

**Keywords:** Y-chromosome DNA, MSY haplotypes, horse diversity, Italian native horses, local breeds

## Abstract

Horse domestication and breed selection processes have profoundly influenced the development and transformation of human society and civilization over time. Therefore, their origin and history have always attracted much attention. In Italy, several local breeds have won prestigious awards thanks to their unique traits and socio-cultural peculiarities. Here, for the first time, we report the genetic variation of three loci of the male-specific region of the Y chromosome (MSY) of four local breeds and another one (Lipizzan, UNESCO) well-represented in the Italian Peninsula. The analysis also includes data from three Sardinian breeds and another forty-eight Eurasian and Mediterranean horse breeds retrieved from GenBank for comparison. Three haplotypes (HT1, HT2, and HT3) were found in Italian stallions, with different spatial distributions between breeds. HT1 (the ancestral haplotype) was frequent, especially in Bardigiano and Monterufolino, HT2 (Neapolitan/Oriental wave) was found in almost all local breeds, and HT3 (Thoroughbred wave) was detected in Maremmano and two Sardinian breeds (Sardinian Anglo-Arab and Sarcidano). This differential distribution is due to three paternal introgressions of imported stallions from foreign countries to improve local herds; however, further genetic analyses are essential to reconstruct the genetic history of native horse breeds, evaluate the impact of selection events, and enable conservation strategies.

## 1. Introduction

In the last two decades, the interest in preserving and improving specific traits in livestock has increased considerably, and several conservation strategies have been implemented in Europe and worldwide. Both at a global and local level, the development of genetic resources has always been a more multifaceted and dynamic process strictly tied to human history.

In farm animals, the evolutionary forces of mutations, breeding and genetic selection, adaptation, isolation, and genetic drift have generated a wide diversity among and between local populations. Also, Italy possesses animals that have inherited great genetic diversity, including several native horse breeds that have reached important national and international recognition thanks to their cultural characteristics and productive traits (www.fao.org/dad-is/, accessed on 22 March 2023).

Even if the whole genomic approaches are now opening up new clues on livestock complexity and admixture, the mitochondrial DNA (mtDNA) and the male-specific region of the Y chromosome (MSY) continue to be extremely important sources of information because of their unique pattern of inheritance, as demonstrated in different species such as sheep and goat [1,2,3,4], cattle [5,6,7,8], and horse [9,10,11,12,13,14,15,16,17,18,19,20,21,22,23,24,25,26,27,28,29,30].

As already mentioned, these markers have a key role in livestock breeding since they allow for evaluating the genetic variation within and among breed lines and could contribute to different fields (i.e., linkage mapping, paternity tests, genome-assisted selection, analysis of genetic diversity within breeds, relationships between breeds, etc.).

In our previous survey [31], we provided the first comprehensive reassessment of the mitochondrial genetic relationships among ten native Italian breeds, representing the most important riding horses and ponies in Italy. We also included 36 Arabian horses as an external reference group to evaluate the genetic effects of its intensive use for the improvement of some local breeds. The high haplotype diversity, shown together with the presence of all domestic mtDNA lineages, demonstrated a widespread mitochondrial variability in Italy, which probably reflects the genetic legacy of ancestral local mares. The only significant genetic differentiation was attributable to geographically isolated contexts (i.e., Sardinian breeds). This effect was also confirmed when broadening the analysis to include other Eurasian horses, highlighting the intermediate position of Italian stocks, which is close to most of the other Mediterranean breeds.

Since Y-chromosome genotypes are only transmitted by stallions to male individuals, they can be analyzed in a more reduced sample than mtDNA. Moreover, the MSY haplotypes’ distribution was strongly influenced by the breeding schemes used since domestication and focused on the refinement of local stocks.

The stallions used for breed improvement were often imported from foreign countries, and their heritage was amplified by their sons and grandsons. This caused the establishment of a “sire line”. Due to the intensive selection of stallions, the predominance of a few sire lines is the rule over the exemption in horse breeds [20]. The Thoroughbreds have a well-documented closed studbook which allows a binding example of this sex bias in horse breeding since the tail-male lineages are linked to only three retained foundation sires: Darley Arabian, Byerley Turk, and Godolphin Arabian [32].

Nevertheless, here, we report the MSY variation of four local Italian breeds together with Lipizzan, a well-represented breed in Italy, and compare them to three Sardinian breeds (Giara, Sarcidano, and Sardinian Anglo-Arab) from [23] and forty-eight other Eurasian and Mediterranean horse breeds retrieved from GenBank in order to rewrite the genetic history of native Italian horses from a male perspective. Recently, new achievements have been made in the analysis of horse sire lines, deepening the molecular phylogeny of the main breeds worldwide and giving important insights into the breeding history [18,33,34,35,36,37]. However, we performed a preliminary analysis focused on the first Y-chromosomal loci reported for the horse [12] by considering the Italian landscape. Therefore, improving knowledge about the genetic structure of the extant Italian breeds in a wider context of both maternal and paternal sides will be essential to further reconstruct their genetic history, evaluate the impact of selection events, and enable conservation strategies.

## 2. Materials and Methods

The present study involves samples derived from the following Italian horse breeds Bardigiano (N = 16), Lipizzan (N = 15), Maremmano (N = 10), Monterufolino (N = 9), and Murgese (N = 10), here briefly described [38,39].

Bardigiano is a brachomorphic horse, which is small and with a robust constitution. It is known as the “mountain breed”, as since ancient times, it has been mainly employed as a pack horse in the Apennines around Parma, Modena, and Reggio Emilia. It is believed to derive from horses that Belgian Gauls rode during their invasions of Rome, and the breed was officially recognized in 1977, with the aim of preserving and recording its genetic heritage.

Lipizzan is a meso-dolichomorph horse with a grey coat. The Italian nucleus (Lipizzan State Stud; ASCAL) includes six male bloodlines (Maestoso, Pluto, Conversano, Favory, Napolitano, and Siglavy) and the original female bloodlines. In 2022, UNESCO recognized this breed as part of the intangible cultural heritage of humanity.

Maremmano is an autochthonous Italian horse breed, probably descending from the native horses of the Etruscans, which represents an important cultural resource. The genealogies of all stallions were completely investigated, and four major male-line founders were identified: the Otello, bay-brown, born in 1927; Salernitano (a Maremmano-like breed originating in the Southern Tyrrhenian cost); the Ussero, with unknown birth data and coat colour; the Thoroughbred Aiace (bay born in 1926 from Darley Arabian); the Thoroughbred Ingres (bay born in 1946 from Godolphin Arabian).

Today, the Monterufolino pony is represented by a few halfbreeds or ponies of uncertain genetic origin. In 1913, the Count Ugolino della Gherardesca of Bolgheri bought the Monterufoli estate, where there were subjects adapted to every form of roughness. Then, there was a reduction in the population size due to the growing mechanization of agriculture, and in the 1980s, there was a recovery of the residual subjects. In 2011, Pomarance municipality adopted the last herd of free-roaming horses living in the hills of Monterufoli and entrusted them to the “Cavallino di Monterufoli group”.

The Murgese is a mesomorph horse developed in the Murge region of Apulia in Italy during Spanish rule. Between 1400 and 1800, it was influenced by several valuable Neapolitan, Andalusian, Arab, and Middle Eastern stallions, and three founding stallions were identified: Granduca (1919), Nerone (1924), and Araldo delle Murge (1928).

Giara is a small horse that originated in the Giara plateau, central-southern Sardinia, and is presumed to descend from the Numidian stock (North African horses, perhaps Barb) brought to the island by the Carthaginians before the Roman era. However, there is some disagreement about the origin of this horse breed and its arrival on the plateau. It was probably brought to the island by the Phoenicians (1500–300 BC) and has always lived in the wild. Currently, the Giara horse is one of the best-known native animals of Sardinia, and it has evolved into a hardy animal due to strong winds, wide temperature swings, and short snowfalls.

Sarcidano is a breed of semi-feral horses that originated in the Oristano province of Sardinia. Its origins may be traced back to the Iberian horse breed around 25,000 years ago. In 1999, Laconi municipality purchased a herd of 15 horses from a private owner, totaling about 100 horses in 2006. A second herd, privately owned, lives in the same region of Sarcidano plateau, while a third group is owned by the Horse Breeding Institute of Sardinia in the center of the northern part of Sardinia.

Sardinian Anglo-Arab horse breed originated from the admixture of indigenous Sardinian mares with oriental-bred stallions and later with French-bred Anglo-Arabian stallions. In 1915, Captain Grattarola (the Director of the Ozieri Remount Station, founded in 1874 to provide mounts for the Italian Army’s cavalry) resumed the operation by crossing the 600 finest available mares with Oriental Thoroughbred stallions, obtained directly from the Bedouin desert tribes. Since the Sardinian indigenous horses were crossed with Arabians and eventually Thoroughbreds, the local foundation bloodstock was established, thus creating the Sardinian Anglo-Arab breed, whose descendants also contributed to founding the so-called Italian Saddle Horse or Sella Italiano.

A total of 1.2 mL of peripheral blood samples was collected from five horse breeds derived from the Italian Peninsula for a total of 60 stallions (Appendix A).

When available, we also collected genealogical data reported in Studbooks (Bardigiano, Lipizzan, Maremmano, and Murgese) or Former Anagraphic Registers (Monterufolino) in order to avoid directly paternally related samples.

Total DNA was extracted from blood samples by automated extraction using the MagCore^®^ Automated Nucleic Acid Extractor, following the provided protocol.

Three polymorphic sites of the MSY (YE3, YE17, and YXX) were analyzed for the 60 stallions belonging to the five breeds as previously reported [12,23] (Appendix A). PCR amplification and Sanger sequencing were carried out as described in [23]. Precisely, PCR amplifications were carried out by using three pairs of oligonucleotides for each polymorphic site: the forward primer (for) 5′-CCCTCTGCTGAGCATCTAGG-3′ and the reverse (rev) 5′-GGCTTAGGCCACTGATGGTA-3′ for YE3; for 5′-GGCCTAAGTTGTTCGCAGAG-3′ and rev 5′-TGACTGGTGGTGTCCAGTGT-3′ for YE17; for 5′-CCTCCGGCCTTTATGTCTTAG-3′ and rev 5′-TTGGGCTGCAGTATACAACG-3′ for YXX. PCR reactions contained 1× Buffer GoTaq, 2.5 mM of each dNTP, 0.3 µM of each primer, 0.02 U/µL of GoTaq DNA polymerase (Promega Corporation; Madison, WI, USA), 30 ng of genomic DNA, and H_2_O to a final volume of 25 µL. PCR amplification was carried out as follows: 95 °C for 3 min, followed by 35 cycles of 95 °C for 30 s, 62 °C for 30 s, 72 °C for 1 min, and then 72 °C for 5 min. The PCR fragments were purified using exonuclease I and alkaline phosphatase (ExoSAP-IT enzymatic system-USB Corporation, Cleveland, OH, USA) and subsequently Sanger-sequenced with the forward primer 5′-GCCAAACTACTCACCAGAAA-3′ (YE3), 5′-GATTACCTCCTGGGACAAC-3′ (YE17), and 5′-TAAAAACCTGTGGAAGGATAA-3′ (YXX). Sequences were, respectively, aligned to the *Equus caballus* haplotype HT1 Y-chromosome locus YE3 (JX646942.1), locus YE17 (JX646950.1), and locus YXX (JX647030.1) for the haplotype annotation. Our dataset was implemented with 34 previously published Y-chromosome haplotypes derived from 3 Sardinian native breeds (Giara, Sarcidano, and Sardinian Anglo-Arab) [23] and compared to 48 other Eurasian and Mediterranean breeds (for a total of 672 stallions), which are publicly available. Two median-joining trees were built to evaluate the evolutionary relationships among samples using Network software v.10.0 (www.fluxus-engineering.com), and Principal Component Analyses (PCA) were performed using Excel software implemented by GenAlEx v.6.41.

## 3. Results and Discussion

### 3.1. Y-Chromosome Haplotypes in the Italian Breeds

The present study involves a total of 60 specimens belonging to the following horse breeds: Bardigiano (N = 16), Lipizzan (N = 15), Maremmano (N = 10), Monterufolino (N = 9), and Murgese (N = 10). The overall alignment of MSY sequences to the references JX646942.1 (YE3, nucleotide positions (nps) 10,592–11,330), JX646950.1 (YE17, nps 1240–1410), and JX647030.1 (YXXX, nps 25,342–25,480) showed three different haplotypes (HT1, HT2, and HT3) [12,23,29] among the six haplotypes previously identified [12,14,27] (Appendix A). No mutations were detected at the locus YXX. By including the three Sardinian breeds previously analyzed (Giara, Sarcidano, and Sardinian Anglo-Arab), the most represented haplotype was HT2 (51%), followed by HT1 (35%) and HT3 (14%) in a total of 94 sequences (Appendix A; Figure 1).

The Median-joining Network analysis showed a different genetic variability among the 60 Italian male horses from this study (Appendix A) and 34 Sardinian samples from [23] (Appendix A) (Figure 1A). Except for Sarcidano, which presents all three haplotypes, the other breeds belong to only one or two Y-chromosomal haplotypes. Despite the difference in the number of samples, HT1, which is considered the ancestral haplotype [12,29], is particularly frequent in Bardigiano, Monterufolino, and Sarcidano and totally absent in Maremmano, Murgese, and Sardinian Anglo-Arab. HT2, a marker of the Neapolitan/Oriental wave, is represented in all breeds apart from Bardigiano, also reflecting its worldwide distribution [12,29] in this microgeographic context, while HT3, typifying the Thoroughbred wave, was found in only three Italian breeds (Maremmano, Sarcidano, and Sardinian Anglo-Arab).

In order to summarize the information held in these MSY haplotypes, we performed a principal component analysis (PCA, Figure 1B), where a clear separation could be observed between Sarcidano and the other Italian breeds due to its variability in terms of HTs. The other breeds are each pushed out from different haplotypes: HT1 for Bardigiano and Monterufolino; HT3 for Maremmano and Sardinian Anglo-Arab; and HT2 for Giara, Lipizzan, and Murgese (Figure 1B).

In general, the MSY haplotypes’ distribution has been strongly influenced by the breeding schemes used since domestication and focused on the refinement of local stocks. The analysis of spatial frequencies of MSY haplotypes in Italy displayed an interesting geographical distribution (Figure 1C). HT1 seems to be preferentially distributed in Northern Italy (Bardigiano and Monterufolino) and Sardinia (Giara and Sarcidano), with three samples also belonging to Lipizzan, which is a breed coming from Lipica (Slovenia) anyway. The most frequent was HT2, whose distribution is influenced by the strong spreading of Neapolitan and Oriental stallions from the Middle East [37]. A different observation must be noted for HT3, typical of Thoroughbred derived from Oriental lineages and distributed across many warmblood horses, which is mostly frequent in Maremmano and Sardinian Anglo-Arab, thus confirming their involvement in breeding programs focused on the improvement of sport aptitudes. Indeed, Maremmano samples belong to the paternal lines Otello, Ussero, Godolphin Arabian, and Darley Arabian. Otello and Ussero are considered native stallions, while Godolphin Arabian and Darley Arabian are the Thoroughbred ancestors, which, respectively, gave rise to Ingres and Aiace sire lines in the Maremmano breed [40]. Among these four lineages, only Darley Arabian presented the HT3 (Thoroughbred wave), while the others belonged to HT2. This could be explained by the intensive use of Darley Arabian over the last 175 years, which is now responsible for 95% of paternal lineages in the modern Thoroughbred population [32]. This finding was also confirmed by Wallner and colleagues [12], who stated that the mutation leading from HT2 to HT3 occurred in the germline of the racehorse Eclipse [12], the great-great-grandson of Darley Arabian [32].

### 3.2. The Italian Y-Chromosome Haplotypes in the Eurasian and Mediterranean Contexts

To build a frame of the Italian MSY haplotypes from this study and [23] in the Eurasian and Mediterranean contexts, we analyzed a total of 56 breeds (753 stallions) by including data from GenBank (Appendix A). Among the haplotypes identified in this study, the highest frequency was recorded for HT1 (44%), followed by HT2 (35%) and HT3 (12%), while the other HTs were retrieved only in foreign breeds: HT4 (2%) in Icelandic [12,14] and Kushum [27] horses; HT5 (2%) in the Norwegian Fjord Horse [12,14]; and HT6 (5%) typical of Shetland and Tiger ponies [12,14] (Appendix A). The high frequencies of HT1 and HT2 seem to be due to the importation of Arabian horses into Central Europe through the so-called Oriental wave [29], while the genetic relationships of Maremmano and Sardinian Anglo-Arab with breeds strongly improved with the English Thoroughbred were here confirmed by the detection of HT3 in these modern horses [12,23,29].

The comparison between our dataset and data from other Eurasian and Mediterranean horse breeds, through a second PCA based on their genetic distances, highlighted some peculiarities (Figure 2).

The plot points out the relatedness of Bardigiano and Monterufolino with breeds from Central and Northern Europe due to the high frequencies of HT1, while Murgese clustered substantially with breeds from Eastern Eurasia, thus confirming results obtained from the mtDNA for the female counterpart [31].

The peculiar localization of Bardigiano could be explained by its origins, which seem to be traced back to the introduction of stallions from different breeds. A further interesting position is occupied by Murgese, which is an ancient breed that originated during the Spanish domination (16th–18th centuries) in Apulia and is closely related to the old Neapolitan horse. The PCA position of the Lipizzan horses from this study, which belongs to the breeding farm of Monterotondo, did not overlap with Lipizzan stallions analyzed by [12,14], although they present the same haplotypes but at different frequencies.

## 4. Conclusions

In this paper, three Y-chromosome loci of 60 stallions from informative areas and breeds present in the Italian Peninsula were analyzed together with 34 samples belonging to the Island of Sardinia. From a male perspective, most of the Italian Y-chromosome gene pool can be traced to three main haplotypes (HT1, HT2, and HT3) due to three paternal introgressions (ancestral haplotype, Neapolitan/Oriental wave, and Thoroughbred wave, respectively) of imported stallions from foreign countries to improve local herds, thus causing the replacement of autochthonous Y chromosomes and a loss of genetic diversity. In particular, the Maremmano breed showed a noticeable frequency of HT3, recorded in samples belonging to the Aiace sire line, which derives from Darley Arabian Thoroughbred and testifies for his intensive use over the last 175 years for the improvement of local breeds.

These results, compared with other Eurasian and Mediterranean data available from the literature, provided a more comprehensive overview of the MSY variation in Italy relative to previous studies and highlighted a preferential distribution of some haplotypes among different breeds.

Nevertheless, this study represents a preliminary analysis through a fast and easily repeatable protocol that can also be undertaken by breeders’ associations and other stakeholders, but the recent enhancement of our knowledge through the first assembly of equine MSY and the refinement of its phylogeny [37] stimulates further research aiming to individuate unique MSY lineages within indigenous populations in order to identify the national populations, understand the historic improvement of local breeds, evaluate the possible influence of mare lines from Italy on foreign populations, and define conservation priorities.

## Figures and Tables

**Figure 1 genes-14-01602-f001:**
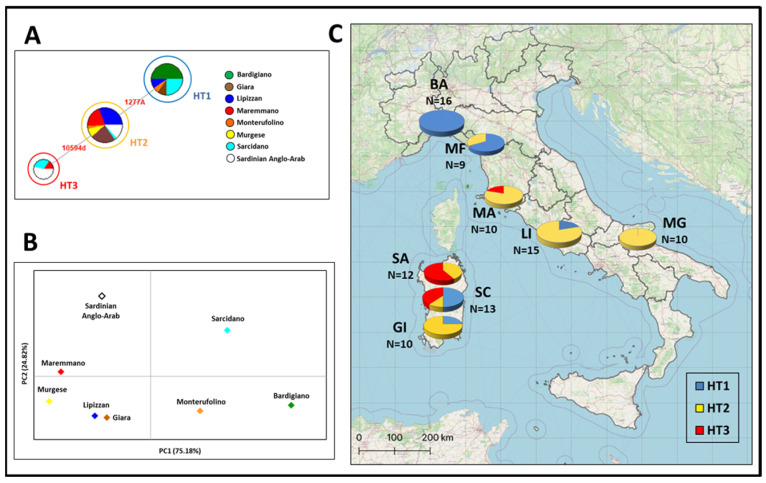
Median-joining Network based on the Y-chromosome haplotypes of eight Italian horse breeds (**A**), where circles are proportional to the observed frequency for each haplotype (HT1: N = 33; HT2: N = 48; HT3: N = 13); PCA plot representing the genetic landscape of Italian breeds based on their genetic distances (**B**); spatial MSY haplotype frequencies of 94 stallions considered here (**C**). Further details are reported in Appendix A.

**Figure 2 genes-14-01602-f002:**
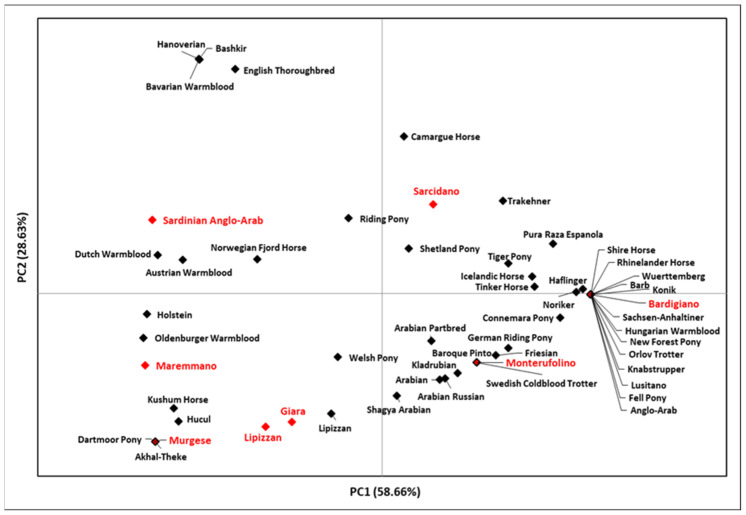
PCA plot representing the genetic landscape of Italian breeds and other Eurasian and Mediterranean breeds based on their genetic distances. The eight breeds analyzed in this study are indicated in red; the indicator is outlined in black when shared with breeds retrieved from GenBank. Further details are reported in Appendix A.

## Data Availability

Data supporting Y-Chromosome haplotypes and reported results can be found in Materials and Methods section.

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
