# Peer review of "Y-Chromosome Haplotype Report among Eight Italian Horse Breeds"

_genes, 2023, doi:10.3390/genes14081602_

Round 1
Reviewer 1 Report (Previous Reviewer 1)
nothing new to add
Author Response
Authors’ response: we would sincerely thank the reviewer for his positive comment.
Reviewer 2 Report (New Reviewer)
I am pleased to have the opportunity to review a paper under the title “Y chromosome haplotype report among eight Italian horse 2 breeds”.
As a result of the review, I have some concerns which I hope should be discussed. The section material and methods with results have to be reorganized for a better understatement of the number of animals used within the study. The results and discussion should be improved. Overall, the manuscript is important to the local genetic resources and might have the potential to be published in a Journal with smaller IF ( Animals?).
Please find below detailed information:
Ln 51-52: Are you sure these particular markers are involved in actions listed within these lines?
Ln 57-62: This part would contribute to the discussion, similar to ln 65-67.
Ln 64-65: Oversimplification. In general mtDNA and male MSY are transmitted mother to – daughter and father to son so why can be analysed in a more reduced sample?
Ln 76-77: “Sardinia breeds form 23”- please provide more information.
Ln 88-89: Please provide the exact number of horses for each of the five breeds involved in the study as referred to in the introduction. This might be organized in a table, with an exact number of samples, the origin of the study and other details. It might be valuable to do a table with a brief description of Italian breeds in general but in my opinion not within the main manuscript.
Ln 141-143 please place firstly in the material and methods section.
Ln 146-146: Ethic statement should be in the last acapit of material and methods, or in line with mdpi policy
Ln 151-152: the methods should be described in a more detailed manner.
Have you done pedigree research? if so how do you state for which breed? and were the results?
Author Response
Please see the attachment

This manuscript is a resubmission of an earlier submission. The following is a list of the peer review reports and author responses from that submission.
Round 1
Reviewer 1 Report
The manuscript is very well written and easy to understand. However, I do not understand why you limited the study to only three loci that were used ten years ago by Wallner et al in 2013. Much more has been done since, even from the same group and I do not understand why this is not even mentioned. The resolution of Figure 1 would substantially improve if key markers (out of the probably several thousands?) defining relevant haplotypes (see https://www.nature.com/articles/s41598-019-42640-w , https://www.mdpi.com/2073-4425/13/2/229 , https://www.mdpi.com/2076-2615/12/24/3508 or https://www.mdpi.com/2076-2615/12/19/2579 just to mention a small selection of the most recent publications on that kind of work) would be used to refine the phylogenetic network. The paper would be nice as it is if we were not in 2023 - I highly recommend to consider the achievements that have been made since 2013 and resubmit an improved and more up to date version that will for sure result in a much more refined network and gives more room for solid interpretations and conclusions and might be of more use for readers and follow-up studies.
Figure 1a: resolution in A could be much higher if the latest Y markers instead of only three from Wallner et al., 2013 were used (of which one is even uninformative for the given dataset). Legend for circle sizes or similar to get an idea of numbers (ok, it is mentioned in text, someone else has to decide if it is required in the legend itself)?
Figure 1b and Figure 2: how much variation is explained by each of the components? Remove title in Figure 2.
Supp Figure 1 and Supp Table 2: I have problems with the color codes and the codes connecting the pies in the figure and how they are represented in the table (what about YE3/1107-11315 in the table and how is that shown in the figure, for instance?). In the abstract it is mentioned: “we have compared them to three Sardinian breeds previously published and other 48 Eurasian and Mediterranean horse breeds retrieved from GenBank.” - does it make sense to summarise “GenBank” samples vs. the others? Based on what were they selected?But again I think the problem would be solved by simply refining the network with more up to date Y markers - currently I do not see much value in Figure S1 compared to Fig 1A. I do not get columns C, E and F in the table and why it is incomplete (especially the difference from E and F? And how does column C help the reader, where else is that code present?).
Line 142: microgeografic context —> microgeographic
Line 217: of equine MSY —> reference?
The manuscript is very well written and easy to understand. Thanks for taking the time to avoid typos etc.
Author Response
The manuscript is very well written and easy to understand. However, I do not understand why you limited the study to only three loci that were used ten years ago by Wallner et al in 2013. Much more has been done since, even from the same group and I do not understand why this is not even mentioned. The resolution of Figure 1 would substantially improve if key markers (out of the probably several thousands?) defining relevant haplotypes (see https://www.nature.com/articles/s41598-019-42640-w , https://www.mdpi.com/2073-4425/13/2/229 , https://www.mdpi.com/2076-2615/12/24/3508 or https://www.mdpi.com/2076-2615/12/19/2579 just to mention a small selection of the most recent publications on that kind of work) would be used to refine the phylogenetic network. The paper would be nice as it is if we were not in 2023 - I highly recommend to consider the achievements that have been made since 2013 and resubmit an improved and more up to date version that will for sure result in a much more refined network and gives more room for solid interpretations and conclusions and might be of more use for readers and follow-up studies.
- Authors’ response: we would like to thank the reviewer for this comment. We completely agree with him and we have included more updated papers in the Introduction. Nevertheless, our manuscript is a brief report and our aim was to use a fast molecular approach which could have been useful also for questions linked to the selection. To date, there are not Y-chromosomal data about some horse breeds from Italy, thus we would perform a preliminary analysis and then move to higher levels of resolution, by focusing on more genetic markers.
Figure 1a: resolution in A could be much higher if the latest Y markers instead of only three from Wallner et al., 2013 were used (of which one is even uninformative for the given dataset). Legend for circle sizes or similar to get an idea of numbers (ok, it is mentioned in text, someone else has to decide if it is required in the legend itself)?
- Authors’ response: we would like to thank the reviewer for this suggestion. We have included the sample number information about each HT in the legend of Figure 1A and modified it as follows: “Figure 1. Median-Joining Network based on the Y-chromosome haplotypes of eight Italian horse breeds (A), where circles are proportional to the number of samples for each haplotype (HT1: N= 29; HT2: N= 39; HT3: N= 13); PCA plot representing the genetic landscape of Italian breeds based on their genetic distances (B); spatial MSY haplotype frequencies of 81 stallions here considered (C). Further details are reported in Tables S1”.
Figure 1b and Figure 2: how much variation is explained by each of the components? Remove title in Figure 2.
- Authors’ response: we would thank the reviewer for the observations. We have modified the figures by adding the variation for each PC and removed the title from Fig. 2.
Supp Figure 1 and Supp Table 2: I have problems with the color codes and the codes connecting the pies in the figure and how they are represented in the table (what about YE3/1107-11315 in the table and how is that shown in the figure, for instance?).
- Authors’ response: we would like to thank the reviewer for this comment. In the column F of Suppl. Tab. S2 is reported the HT code used for buildin the network, thus there is a correspondance between the columns E and F (i.e. the variant “YE3/1107-11315 20SNPs-4indels” reported by Wallner et al. 2013 is “20S 4id” in our network of Fig. S1).
In the abstract it is mentioned: “we have compared them to three Sardinian breeds previously published and other 48 Eurasian and Mediterranean horse breeds retrieved from GenBank.” - does it make sense to summarise “GenBank” samples vs. the others? Based on what were they selected?But again I think the problem would be solved by simply refining the network with more up to date Y markers - currently I do not see much value in Figure S1 compared to Fig 1A.
- Authors’ response: we would like to thank the reviewer for this comment. In the abstract we wanted to specify that we have considered the other Eurasian and Mediterranean horse breeds analysed for the same loci. The network analysis would certainly be more accurated by including the other Y-chromosomal markers; however, as it is a brief report, we wanted to perform a preliminary analysis for these equine breeds from Italy. Then, we aim to move to higher levels of resolution.
I do not get columns C, E and F in the table and why it is incomplete (especially the difference from E and F? And how does column C help the reader, where else is that code present?).
- Authors’ response: we would like to thank the reviewer for this comment. Column F is incomplete when the sample is identical to reference sequence in that locus; anyway, we have filled in the empty cells. The difference between columns E and F is only that in F we have reported the code used for building the network, and if the reader wants to know more about these codes, he can find the variants previously published by Wallner et al. 2013 in column E. On the other hand, the column C does not supply important information to the reader, as they are only the codes used to built our network. Thus, we have removed this column from the Suppl. Tab. S2.
Line 142: microgeografic context —> microgeographic
- Authors’ response: we have replaced “microgeografic” with “microgeographic”.
Line 217: of equine MSY —> reference?
- Authors’ response: we have added the reference.
Comments on the Quality of English Language
The manuscript is very well written and easy to understand. Thanks for taking the time to avoid typos etc.
Authors’ response: we would like to thank the reviewer for this comment.
Reviewer 2 Report
This paper is a brief report of Y-chromosomal haplotypes in Italian horses. The authors sampled stallions of four native Italian breeds (Bardigiano, Maremmano, Monterufolino, and Murgese), and Italian Lipizzan horses (N=47) and screened the samples for three variable nucleotide sites present in Y-chromosome by PCR and Sanger-sequencing to form Y-chromosome haplotypes. Then, they added previously published haplotypes from native Sardinian breeds (Giara, Sarcidano and Sardinian Anglo-Arab, N = 34) to their data and compared these Italian haplotypes to haplotypes of 48 Eurasian and Mediterranean breeds obtained from GenBank (N=672). The newly studied Italian breeds were found to possess three (HT1, HT2 and HT3) of the previously detected six haplotypes. All three haplotype were found in Sarcidano, whereas the other breeds had only one or two haplotypes. Haplotype HT1, considered as the ancestral haplotype was frequent in Bardigiano, Monterufolino and Sarcidano and absent in Maremmano, Murgese, and Sardinian Anglo-Arab and geographically most common in northern Italy. HT2, a marker of the Neapolitan/Oriental wave, was most common and detected in all breeds except in Bardigiano and HT3, connected with the Thoroughbred wave, was found in Maremmano, Sarcidano and Sardinian Anglo-Arab. When the haplotype frequencies in Italian breeds were compared to that in other breeds by PCA, Bardigiano and Monterufolino were grouped with breeds from Central and Northern Europe, due to the high frequencies of HT1, while Murgese, and to some extent also Maremmano, grouped with breeds from Eastern Eurasia.
Overall, this is a nice report that is likely especially useful for breeders of the Italian native breeds. There are only some issues that I would like to point out, mostly minor. My main concern is on the use of PCA, which I think is not a very good way to show the relatedness of the breeds, as it is based on only three variable nucleotide sites in Y-chromosome forming the haplotypes and their frequencies in the samples are likely much affected by sampling bias as the sample sizes per breed are quite small. You could perhaps try a simple phylogenetic tree and/or a haplotype network including all the previously published data? At least the haplotype network should be quite informative.
Minor issues:
Introduction
- You have written (starting line63) that “Y chromosome genotypes are less informative than mtDNA haplotypes, as they are transmitted by stallions to only male individuals and show a valuable loss of genetic variation throughout the history.” This is a bit odd sentence, as I don’t see how uniparental inheritance of Y differs from uniparental inheritance of mtDNA leading to less information? It is clear that there is less variation in Y than in mtDNA resulting from biology of the species and the breeding schemes used since domestication, but that knowledge I think is informative. Further, how loss of genetic variation in Y can be valuable?
Methods:
- You should briefly describe the study breeds, as most readers are not likely familiar with them.
- Please give numbers of samples per breed also in the main text
- You should add accession numbers for sequences you used from GenBank (at least to the supplementary table) and submit your sequences to GenBank and add their accession numbers as well.
Although overall this is quite well written, English could still be carefully checked as there are some odd choises of words and wording.
Author Response
This paper is a brief report of Y-chromosomal haplotypes in Italian horses. The authors sampled stallions of four native Italian breeds (Bardigiano, Maremmano, Monterufolino, and Murgese), and Italian Lipizzan horses (N=47) and screened the samples for three variable nucleotide sites present in Y-chromosome by PCR and Sanger-sequencing to form Y-chromosome haplotypes. Then, they added previously published haplotypes from native Sardinian breeds (Giara, Sarcidano and Sardinian Anglo-Arab, N = 34) to their data and compared these Italian haplotypes to haplotypes of 48 Eurasian and Mediterranean breeds obtained from GenBank (N=672). The newly studied Italian breeds were found to possess three (HT1, HT2 and HT3) of the previously detected six haplotypes. All three haplotype were found in Sarcidano, whereas the other breeds had only one or two haplotypes. Haplotype HT1, considered as the ancestral haplotype was frequent in Bardigiano, Monterufolino and Sarcidano and absent in Maremmano, Murgese, and Sardinian Anglo-Arab and geographically most common in northern Italy. HT2, a marker of the Neapolitan/Oriental wave, was most common and detected in all breeds except in Bardigiano and HT3, connected with the Thoroughbred wave, was found in Maremmano, Sarcidano and Sardinian Anglo-Arab. When the haplotype frequencies in Italian breeds were compared to that in other breeds by PCA, Bardigiano and Monterufolino were grouped with breeds from Central and Northern Europe, due to the high frequencies of HT1, while Murgese, and to some extent also Maremmano, grouped with breeds from Eastern Eurasia.
Overall, this is a nice report that is likely especially useful for breeders of the Italian native breeds.
- Authors’ response: we would like to thank the reviewer for this comment.
There are only some issues that I would like to point out, mostly minor. My main concern is on the use of PCA, which I think is not a very good way to show the relatedness of the breeds, as it is based on only three variable nucleotide sites in Y-chromosome forming the haplotypes and their frequencies in the samples are likely much affected by sampling bias as the sample sizes per breed are quite small. You could perhaps try a simple phylogenetic tree and/or a haplotype network including all the previously published data? At least the haplotype network should be quite informative.
- Authors’ response: we would like to thank the reviewer for this suggestion. We agree with reviewer about the choise of PCA by using only these three informative sites, thus we have included two network analyses (Fig. 1A and Fig. S1) trying to give a bit more information on the relatedness of breeds. However, it is a preliminary analysis of these equine breeds from Italy and we aim to move to higher levels of resolution with further studies.
Minor issues:
Introduction
- You have written (starting line63) that “Y chromosome genotypes are less informative than mtDNA haplotypes, as they are transmitted by stallions to only male individuals and show a valuable loss of genetic variation throughout the history.” This is a bit odd sentence, as I don’t see how uniparental inheritance of Y differs from uniparental inheritance of mtDNA leading to less information? It is clear that there is less variation in Y than in mtDNA resulting from biology of the species and the breeding schemes used since domestication, but that knowledge I think is informative. Further, how loss of genetic variation in Y can be valuable?
- Authors’ response: we would like to thank the reviewer for this suggestion. We have modified the text as follows: “Since Y chromosome genotypes are transmitted by stallions to only male individuals, they can be analysed in a more reduced sample than mtDNA. Moreover, the MSY haplotypes show a valuable loss of genetic variation throughout the history, mainly due to the breeding schemes used since domestication, which were focused on the refinement of local stocks, and based on a small number of males used for breeding and male-mediated cross-breeding”.
Methods:
- You should briefly describe the study breeds, as most readers are not likely familiar with them.
- Authors’ response: we would like to thank the reviewer for this comment. We have improved the manuscript with brief description of breeds here analysed as follows in Materials and Methods section.
- Please give numbers of samples per breed also in the main text
- Authors’ response: we have added numbers of samples per breed also in the main text.
- You should add accession numbers for sequences you used from GenBank (at least to the supplementary table) and submit your sequences to GenBank and add their accession numbers as well.
- Authors’ response: we would like to thank the reviewer for this observation. We have not added the accession numbers from data retrieved from Genbank since they are reported as SNPs and classified in haplotypes. This is why we have not deposited our sequences to GenBank, but we have preferred publishing the SNP and the HT for each horse sample.
Comments on the Quality of English Language
Although overall this is quite well written, English could still be carefully checked as there are some odd choises of words and wording.
Authors’ response: we would like to thank the reviewer for this suggestion. We have checked the entire manuscript.
Round 2
Reviewer 1 Report
thanks for addressing the points. no further comments.